# Ductile Compressive Behavior of Biomedical Alloys

**Christian Affolter [1],\*, Götz Thorwarth [2], Ariyan Arabi-Hashemi [3], Ulrich Müller [4] and Bernhard Weisse [1]**

[1] Lab for Mechanical Systems Engineering, Empa, Swiss Federal Laboratories for Materials Science and Technology, Ueberlandstrasse 129, CH-8600 Dübendorf, Switzerland; Bernhard.Weisse@empa.ch

[2] IMT Masken und Teilungen AG, Im Langacher 46, CH-8606 Greifensee, Switzerland; gthorwarth@imtag.ch

[3] Lab for Advanced Materials Processing, Empa, Swiss Federal Laboratories for Materials Science and Technology, Ueberlandstrasse 129, CH-8600 Dübendorf, Switzerland; ariyan.arabi-hashemi@empa.ch

[4] Lab for Nanoscale Materials Science, Empa, Swiss Federal Laboratories for Materials Science and Technology, Ueberlandstrasse 129, CH-8600 Dübendorf, Switzerland; Ulrich.Mueller@empa.ch

\* Correspondence: Christian.Affolter@empa.ch

**Abstract:** The mechanical properties of ductile metals are generally assessed by means of tensile testing. Compression testing of metal alloys is usually only applied for brittle materials, or if the available specimen size is limited (e.g., in micro indentation). In the present study a previously developed test procedure for compressive testing was applied to determine the elastic properties and the yield curves of different biomedical alloys, such as 316L (two different batches), Ti-6Al-7Nb, and Co-28Cr-6Mo. The results were compared and validated against data from tensile testing. The converted flow curves for true stress vs. logarithmic strain of the compressive samples coincided well up to the yield strength of the tensile samples. The developed compression test method was shown to be reliable and valid, and it can be applied in cases where only small material batches are available, e.g., from additive manufacturing. Nevertheless, a certain yield asymmetry was observed with one of the tested 316L stainless steel alloys and the Co-28Cr-6Mo. Possible hypotheses and explanations for this yield asymmetry are given in the discussion section.

**Keywords:** biomedical alloys; yield asymmetry; compressive testing; flow curve

## 1. Introduction

The origins of the theory of plasticity go back more than 100 years. A complete work can be found in Hill [1], originally published in 1950. The ductile properties of alloys including their flow curve during yielding are commonly determined with tensile testing, which is a well-standardized procedure. Compressive testing may be the appropriate material test for characterizing a ductile alloy, if the data are used to dimension a structure dominantly loaded under compression, or if the sample size and the physical constraints do not allow standard tensile testing. Kulagin et al. [2] applied compressive testing to analyze the effect of severe plastic deformation (SPD) on the microstructure and mechanical properties of a pure titanium. With even further reduced sample size, testing with micro or nano-indentation may be justified, where yield strength and tensile strength can be deduced recursively by fitting with numerical models.

In an earlier project for the development of hard coatings on biomedical alloys, a test procedure was developed by the present authors [3] in accordance to two standards:

- DIN 50106: originally from 1960, revised in 1978 and recently in 2016. It was mainly written for the determination of the compressive strength of brittle (cast) alloys, and it says very little about the allowable type of strain or displacement measurement. The standard is not particularly designed for the measurement of Young's modulus or yield strength [4].

- ASTM E9-09 describes in particular the adequate test setup with respect to axial alignment and parallelism of two hardened bearing blocks, and it addresses the problems related with buckling of slender test samples and 'barrelling' (the non-uniform deformation in the sample's end region due to friction; [5]).

The new test protocol allowed the determination of Young's modulus, yield strength and the entire flow curve (stress–strain curve in the plastic domain), and it was validated in a Round Robin on a 316L (DIN 1.4441 according to [6]) material against standard tensile testing [3]. The compressive test has an additional benefit, because tensile samples tend to undergo necking after a few percent of plastic strain, which will dictate the ultimate tensile strength (when necking has started, the true local stress may still increase, but the evaluated nominal stress drops). The yield curve in compression may evolve to plastic strains far above the strain $A_f$ (ultimate strain at rupture) of a tensile test.

The hard coatings made of diamond-like carbons (*DLC*) developed in a previous project were investigated by micro indentation [7]: A conical imprint in the coating was produced, superimposing plastic strains and high stresses onto the already existing residual stresses from the coating process. The simultaneously created damage in the DLC-coating lead to a high delamination rate at the interface DLC—Substrate and allowed investigation of different types of coating agents and interlayers. The numerical simulation of the micro indentation tests required the actual flow curve of the different substrate materials, in order to predict the residual stresses, the plastic strain, and finally the energy release rates during delamination. As the available material batches were small and the results were used to simulate an indentation experiment (with dominantly compressive and hydrostatic stresses), the tests were performed on relatively small samples in compression, considering the relevant standards.

When tensile or compressive test data are produced, they are generally evaluated as 'nominal stress' vs. 'nominal strain'. These curves have to be converted to true stress vs. logarithmic strain for many state-of-the-art Finite Element (FE) codes such as *Abaqus* from Simulia/3DS [8]. The true stress can be derived as follows:

$$\sigma_{\text{true}} = \sigma_{\text{nom}} (1 + \varepsilon_{\text{nom}}), \tag{1}$$

and the logarithmic strain measure is defined as

$$\varepsilon_{\ln} = \ln(1 + \varepsilon_{\text{nom}}) \tag{2}$$

The conversion of tensile and compression test curves of identical material batches should result in coinciding flow curves at least up to the start of necking in the tensile experiment (at strain $A_g$). Differences may indicate inappropriate test setups or even wrong boundary conditions.

In the present project, different biomedical alloys were tested in compression, and where the size of the material batch allowed it, tensile tests were performed for comparison. The elastic properties could be determined as well as the entire flow curve under tension/compression, and the conversion of the flow curves to true stress vs. logarithmic strain allowed further interpretation and assessment of the materials. Although the applied methodology in this study is not fully new, the yield curves presented and the verification against tensile data are difficult to find in literature and may be of interest to researchers employing numerical simulation. The observed yield asymmetry in two of the studied alloys was further examined and partly explained.

In the test setup used for this study, the influence of friction could be reduced to a minimum, as shown in [3]. Robinson et al. showed in a previous study how the friction coefficient affects the outcome of compression tests on ring samples [9]: The deformation pattern and the change of the inner ring diameter depend strongly on friction, and the results can be used to recursively determine the friction coefficient or material parameters by means of the Finite Element Method (FEM).

## 2. Materials and Methods

### 2.1. Test Method and Mechanical Setup

The method for the compressive tests was in accordance with ASTM E9-09 and DIN 50106, with some specific considerations concerning strain measurement, friction, and evaluation [3]. A setup was chosen, where the two hardened and polished bearing block surfaces can be set to parallel under preload by means of a spherical calotte, but it was an important requirement to fix the block's surface orientation after reaching the preload such that the calotte cannot further rotate under increased load. Two hardened cones were used to increase the working space between the parallel bearing block surfaces such that the clip-on gauge for strain measurement could be placed on the specimen after the cylindrical sample had been aligned in the machine axis and set under preload (cone material: DIN 1.3351; hardened to a Rockwell hardness ≥65 HRC). The chosen setup is shown schematically in Figure 1. The strain measurement was performed symmetrically on two sides of the cylindrical samples with a gauge type "Mini MFA-2" (from *MF Mess- & Feinwerktechnik GmbH*, Velbert, Germany). At a total strain of approx. 4% the transducer was removed and the following deformation was measured via the crossbar displacement to calculate the plastic strain. As described in [3] the load does not change substantially anymore, hence the distortion of the surrounding test setup can be assumed to be almost constant for increased plastic deformation, and the resulting flow curve will not be significantly influenced.

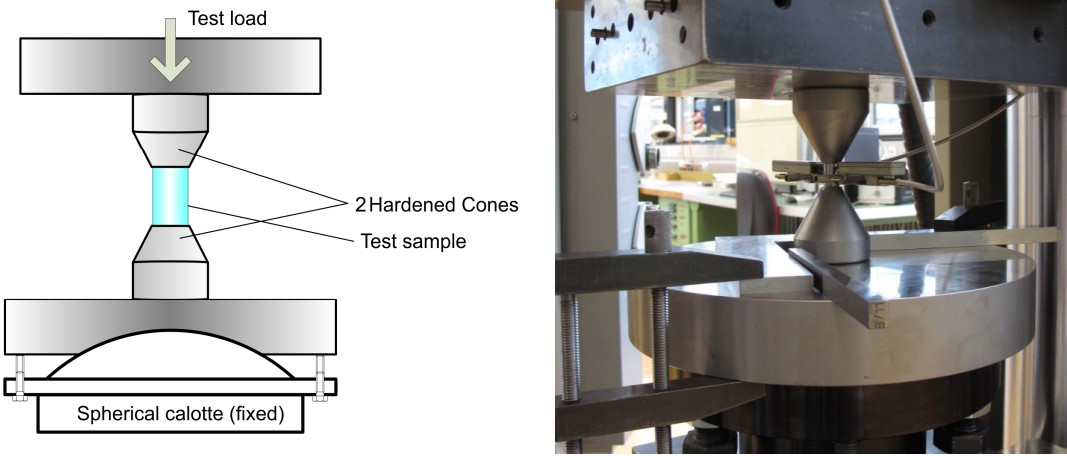

**Figure 1.** Schematic view of the mechanical setup (**left**) and installation with prepared sample in the test machine (**right**), with large plates adjusted and fixed in parallel.

The tensile tests were performed according to DIN EN ISO 6892-1:2009, and the strains were measured with a transducer type *multiXtens* from Zwick (Ulm, Germany) of class 0.5 according to EN ISO 9513. The strain rates for both test procedures were adjusted carefully, such that the strain rate in the elastic domain for tension was similar to the rate in the elastic domain for compression. The strain rates during plastic deformation were set higher by an order of magnitude, but again were comparable for tension and compression

### 2.2. Materials

The materials involved in this experimental study were two types of stainless steel 316L, a titanium alloy and a cobalt chromium molybdenum (CoCrMo), defined by the following specifications:

- A grade AISI 316L stainless steel (DIN 1.4441, implant quality), with further specifications as follows: X2CrNiMo 18-15-3 (ISO 5832-1 UNS S31673, ASTM F138), tensile strength between 930 and 1100 MPa. Polished rod with circular cross section (diameter Ø = 10 mm; tolerance *h6*). The material will simply be called '1.4441'.

- A second batch of medical grade 316L stainless steel (Ø = 18 mm), where the supplier and more detailed specifications are not known, hereafter called '316L'.
- A titanium alloy type Ti-6Al-7Nb, hereafter called 'TAN'.
- A cobalt based Co-28Cr-6Mo, hereafter called 'CCM'.

The material first used for the development of the test procedure and the validation against tensile tests was the 1.4441 stainless steel in implant quality. All tensile and compressive samples had been produced from the same batch of a polished round rod (diameter Ø = 10.00 mm). Out of this rod, and also with the other materials, the following test samples were manufactured:

- Tensile test: according to DIN 50125, type "F 10 × 50" with rod lengths between 330 and 500 mm, or type "B 6 × 30" for shorter samples, where the outer diameter is given by the *M10*-thread. The 316L samples were of type "B 10 × 50" as the raw material was available with larger diameter.
- Compressive test: on the basis of DIN 50,106 and ASTM E9-09 with cylindrical shape, $d_0 = 10$ mm, and $h = 15$ mm which resulted in a $h/d_0$ ratio of 1.5.

All results were evaluated as *nominal stress* vs. *nominal strain*. From each set of tensile and compressive tests, the mean curves were taken and converted to *true stress* vs. *logarithmic strain* (cf. [8]). The conversion of the measured nominal strain provides a shift of the data points in the horizontal direction, in addition to the vertical shift for stress.

## 3. Results

All tests were performed carefully and the data digitally collected for conversion and comparison. Figure 2 shows an overview of all flow curves for plastic yielding, each graph providing the comparison between tensile and compressive tests. For some materials only a limited number of samples was available which does not allow a statistics for the relevant material parameters. For TAN no tensile samples were available at all, hence the test data of the compression tests had to be compared to literature data from Polyakova et al. ('Initial', coarse-grained in [10]).

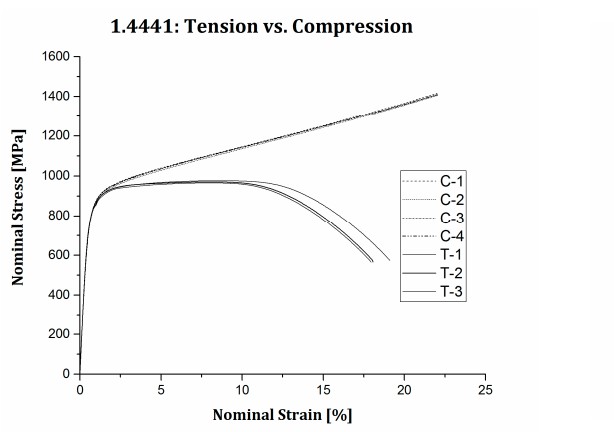 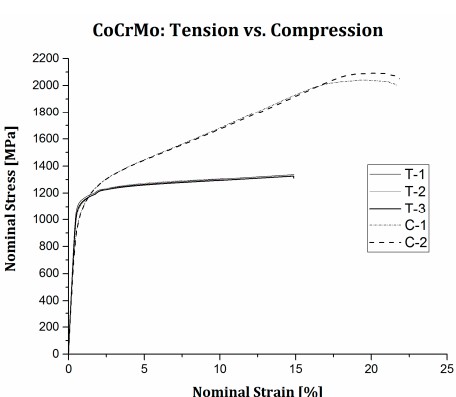

**Figure 2.** *Cont.*

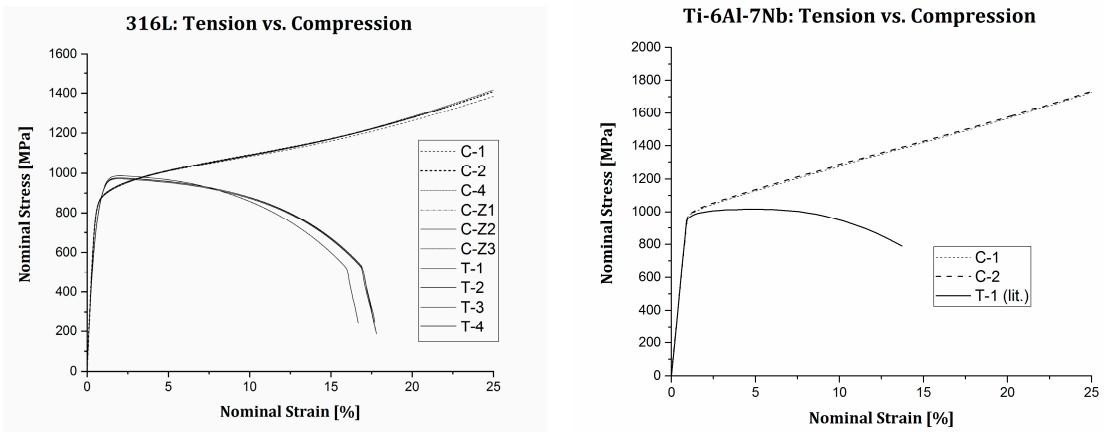

**Figure 2.** All test data, presented as nominal stress vs. nominal strain.

The most relevant material parameters are provided in Table 1. The results for Young's modulus *E* and yield strength $R_{p0,2}$ are in all cases comparable between tension and compression. Only the batches of 316L and CCM show a certain discrepancy in the yield strength, which is already indicated in the test curves. TAN was the only material showing a distinct yield limit. For all other materials, the yield strength is determined at the point of 0.2% permanent plastic deformation ($R_{p0,2}$).

**Table 1.** Overview of material parameters with statistics and comparison of the test results.

| Material | 1.4441 | | 316L | | CCM | | TAN | |
| Type of Testing | Tension | Compr. | Tension | Compr. | Tension | Compr. | Tension | Compr. |
| --- | --- | --- | --- | --- | --- | --- | --- | --- |
| Young's Modulus *E* [MPa] | 178,714 | 174,370 | 179,847 | 179,085 | 228,292 | 220,398 | 105,000 (lit.) | 101,618 |
| $R_{p0,2}$ [MPa] | 778.8 | 777.2 | 760.4 | 842.0 | 1102.3 | 973.9 | 975 | 990.3 |
| $R_m$ [MPa] | 970.7 | n.a. | 977.6 | n.a. | 1323.7 | n.a. | 1020 | n.a. |
| $\varepsilon_{Ult.}$ [%] | 18.4 | n.a. | 17.5 | n.a | 13.4 | n.a. | 13 | n.a |
| No. of samples | 3 | 4 | 4 | 6 | 3 | 2 | (1, lit. [1]) | 2 |

[1] from literature [10].

For a further analysis a mean curve was taken from each set of resulting test curves, and these curves were then converted to *true stress* vs. *logarithmic strain* (according to Equations (1) and (2), see also [8]) which is a common and established procedure for the numerical simulation of ductile (plastic) materials. Figure 3 provides for each material a set of converted curves (continuous) versus the original curves (dashed).

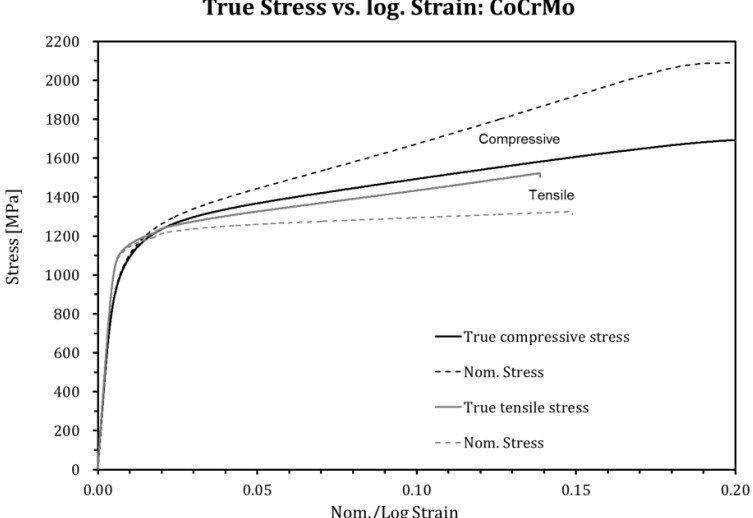

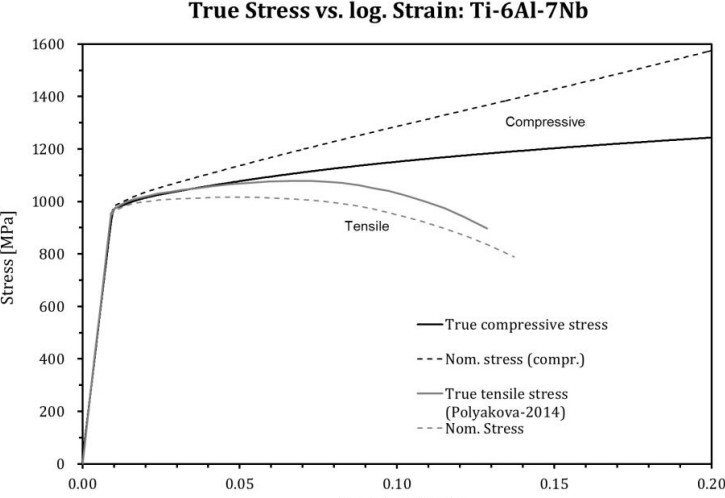

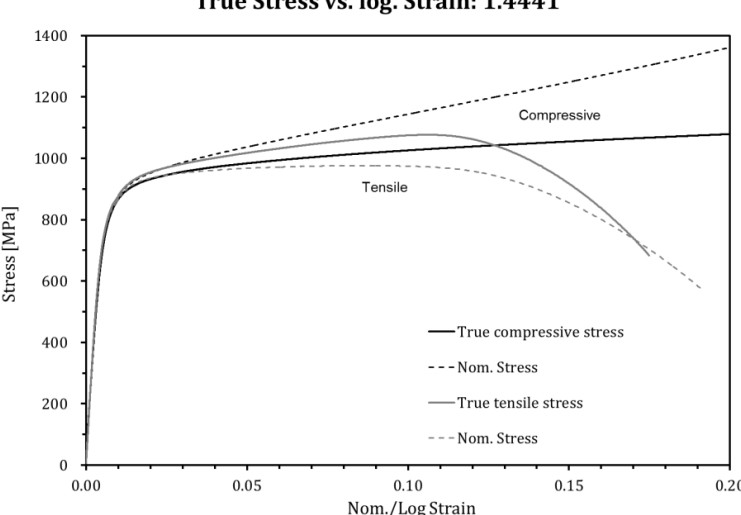

**Figure 3.** *Cont.*

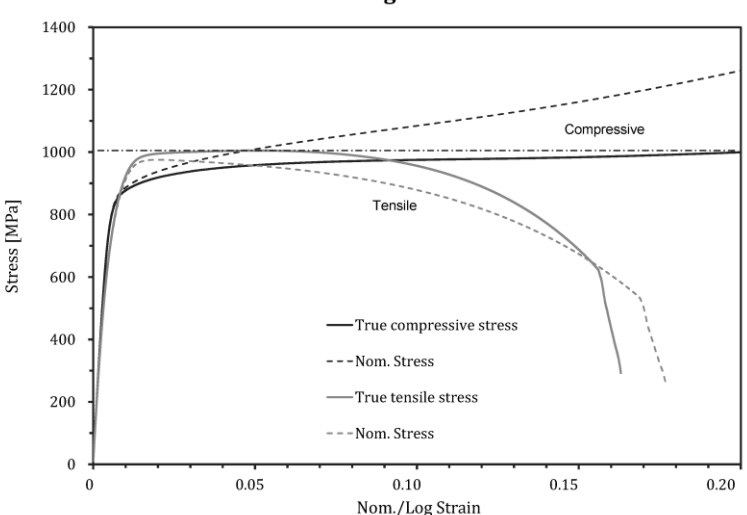

**Figure 3.** Test data displayed as *True Stress* vs. *Logarithmic Strain.*

The only material showing a fast fracture without antecedent necking of the tensile sample is the Co-28Cr-6Mo (CCM), all other materials show necking of the tensile sample during plastic deformation, which leads to the drop of the tensile curve. This drop is also observed in the converted 'true stress' curve due to the purely mathematical conversion process (as the true local stress in the necking area actually could not be measured).

## 4. Discussion

When looking at the converted flow curves for tension and compression in the range between yield strength and tensile strength (true stress vs. log strain), both curves usually run parallel up to the tensile strength $R_m$, where necking of the tensile sample begins (at strain $A_g$). This is especially true for the two materials 1.4441 and CCM; the stainless steel 1.4441 shows slightly higher strength in tension, CCM on the other hand has a higher strength in compression. TAN shows a similar behavior, but both flow curves start rather tangentially from the yield strength and progress with different curvature.

316L showed another behavior than the three previous materials, as necking of the tensile samples seemed to start shortly after reaching the material's yield strength. Both converted flow curves rather had a horizontal envelope in common at approx. 1000 MPa (cf. Figure 3). For this 316L batch even the yield strength was very different in tension and compression, respectively, see also Table 1. This 'yield asymmetry' could also be observed with CCM, though $R_{p0,2}$ of 316L was higher in compression, but for CCM higher in tension. A detailed look at the flow curves of these two materials is shown in Figure 4 in the transition between elastic and plastic behavior (up to 6% total strain).

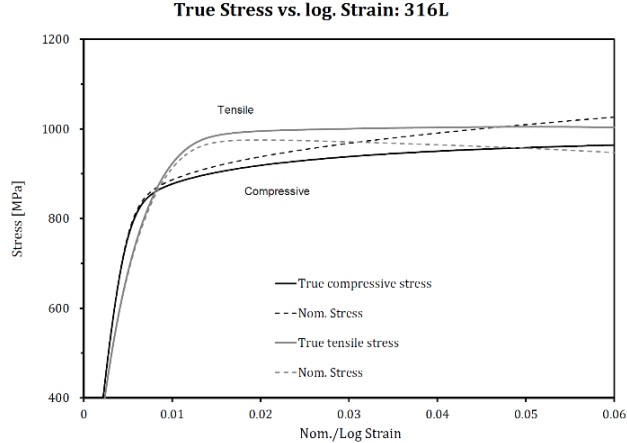

**Figure 4.** Asymmetry in steel 316L (**top**) and Co-28Cr-6Mo (**bottom**).

There are different possible explanations for the yield asymmetry in the 316L stainless steel. Implant quality 316L has preferred a pure austenitic microstructure. Cold working (or machining and mechanical treatment with consequent heating/thermal treatment) may lead to a martensite phase transformation [11]. Martensite is undesirable for implants as it is a highly magnetic phase [12]. Another mechanism which may lead to yield asymmetry is the creation of 'twin' crystals during plastic deformation. The creation of twins is different under tension and compression, which may also lead to an asymmetric behavior during yielding. Phase transformations then occur simultaneously to grain boundary sliding. Magnetism could not be observed in the tested samples, hence martensite phase transformation seems unlikely in the present case.

In order to further reveal the origin of the yield strength asymmetry of 316L, its microstructure was compared with the microstructure of 1.4441 which does not show a yield strength asymmetry. 316L and 1.4441 have the same chemical composition, and both materials show a high degree of deformation twinning in the as-received state, as shown in Figure 5. Alloy 1.4441 has a larger grain size than 316L. The high density of deformation twins in the as-received state is attributed to large plastic deformations of both materials. Due to the difference of grain size and the high amount of deformation twinning it can be expected that both materials have undergone different thermo-mechanical treatments. The reduced grain size in 316L might be the reason for the yield asymmetry favoring yielding under tension.

## a) 316L - as received

## b) 1.4441 - as received

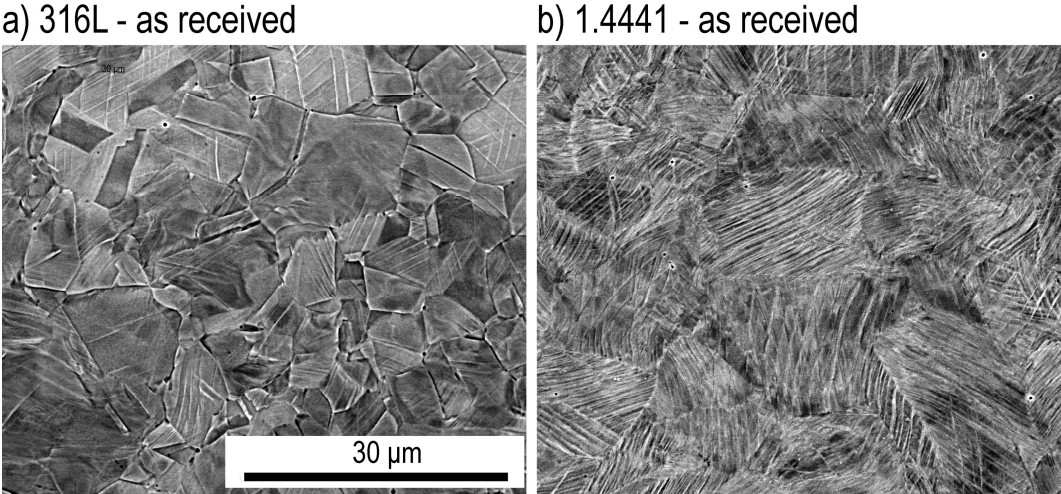

**Figure 5.** Microscopy using backscattered electrons reveals the microstructures in the as-received state of (**a**) 316L and (**b**) 1.4441.

The different thermo-mechanical treatments can cause textures which can both be responsible for the yield asymmetry in 316L. Texture might have an impact on the yield strength asymmetry. In the present case 316L shows a reduced yield strength for tension. XRD and EBSD results show that 316L has a texture where preferentially grains with a <111> orientation are aligned parallel to the loading direction (Figure 6, left). Twinning for <111> orientated grains is easier for tension than for compression [13,14]. Thus the observed texture is in accordance with the observed yield strength asymmetry. However, since both materials 316L and 1.4441 show a <111> texture, texture does not fully explain the observed asymmetry, as it is not present in the case of 1.4441 (Figure 6, right).

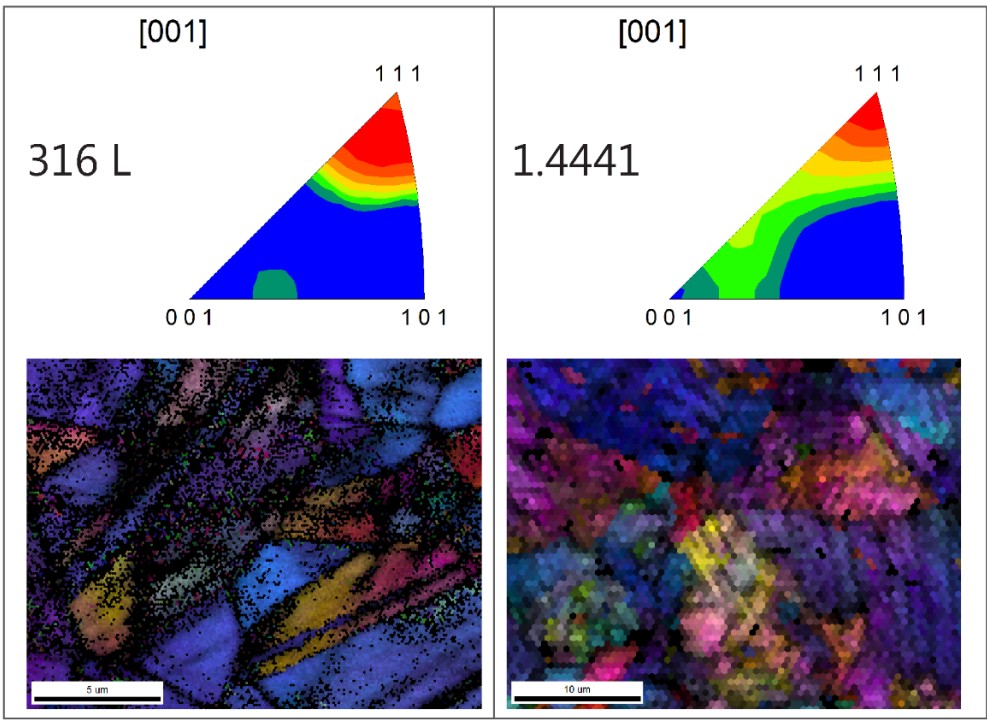

**Figure 6.** Plot (IPF) and EBSD measurement of the two stainless steel materials.

To obtain insights into the microstructural changes of 316L during compression and tension, XRD measurements were done. The measurements were done at a cross-section of the round specimens

which were unstrained and strained to +1.5% and −1.5%. The planes of diffraction of the measurements shown in Figure 7 had their normal axis parallel to the loading direction. In a texture-free sample the (200) peak would have around 50% of the intensity of the (111) peak. Thus in the present case, due to the strong (111) peak, a (111) texture is present as confirmed by the pole figure measurements. The XRD measurements show that the intensity of the (111) peak decreases for tension and increases for compression. This indicates that during tension new twins are formed or existing twins grow. On the other hand during compression it seems that detwinning takes place. That means that existing twins shrink and thus the measured (111) intensity increases.

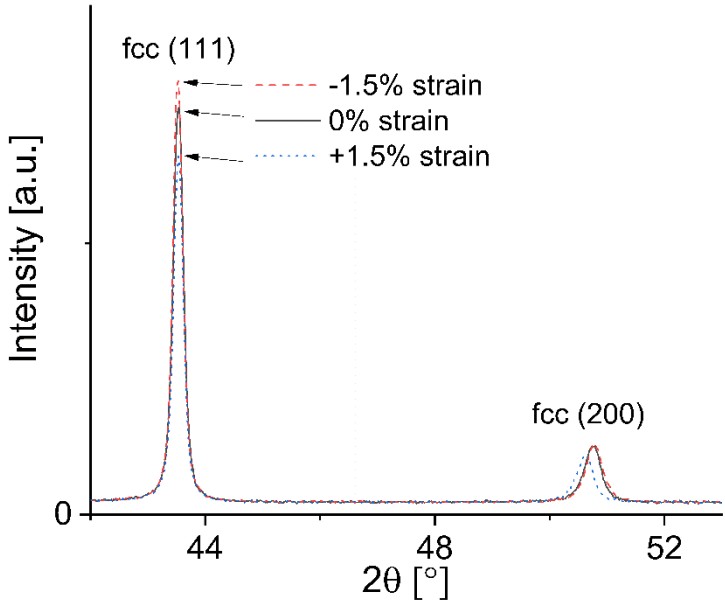

**Figure 7.** XRD measurements of 316L taken at 0% strain, +1.5% and −1.5% strain. The normal vector of the diffraction planes is aligned parallel to the loading direction.

A final explanation may be that the raw material was inhomogeneous. The rods from which the 316L test samples were machined had an outer diameter of 18 mm, the ones for the 1.4441 samples 10 mm. If the tensile and the compressive test specimen had a differing diameter, and if there was a changing grain size or phase composition from the core to the outside of the rods, i.e., a radial inhomogeneity, then machining might lead to a different microstructure through the tested cross sections of tensile and compression samples, respectively, or it might affect (release) internal stresses. However, tensile and compressive samples had identical diameters for both material batches.

## 5. Conclusions

Yield curves were produced for different biomedical alloys by means of tensile and compressive testing, and the curves were cross-validated. The new compression test method ([3]) was shown to be reliable and valid, and it can be applied in cases where only small material batches are available. The occurrence of a yield asymmetry with 316L and CoCrMo could not fully be explained by the microstructural analysis. However, the presented curves are a valuable input for future research and applied R&D (implant development) supported by numerical simulation.

Following recommendations can be given for the implementation of plasticity in finite element modelling: Most of the FE codes will require a flow curve defined as true stress vs. (logarithmic) plastic strain. It is appropriate to implement an averaged curve from tension and compression test data, if both results are available, and if the load case can result in tensile and compressive strains (e.g., in a bending load case). For dominantly compressive load cases it is advantageous to implement a test curve from a compressive test (e.g., simulation of indentation tests). It is particularly inappropriate to implement

the declining right branch of the flow curve which results from necking in the tensile samples. It is simpler and finally more accurate to extend the approximately linear part of the (averaged) flow curve before necking.

**Author Contributions:** C.A. led the study, developed the test procedures and performed the validations also by means of FEM modelling. C.A. also considered the continuum mechanical aspects. G.T. considered practical aspects in the study, and ensured its relevance for practical applications in biomedical engineering and production of hard coatings on metals. G.T. also performed metallographic experiments (EBSD) and helped to analyze the observed yield asymmetries. A.A.-H. performed the microstructural analysis: micrographs, XRD, and EBSD. A.A.-H. investigated the origins of the observed yield asymmetry. U.M. helped to develop the compression test procedure which allows to determine the plastic flow curve, and also supported the numerical modelling in the first phase. B.W. organized the funding and supported the study with theoretical thoughts (continuum mechanics, mechanical testing). All authors have read and agreed to the published version of the manuscript.

**Funding:** The study was partially funded by the Swiss Commission for Technology and Innovation CTI/KTI with the grant No. 8475.1 LSPP-LS.

**Acknowledgments:** The authors express their gratitude to Hans Michel for all the valuable ideas and practical work in the testing lab, and the continuous efforts to improve the test procedures.

**Conflicts of Interest:** The authors declare no conflict of interest.

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
