# Peer review of "Ductile Compressive Behavior of Biomedical Alloys"

_metals, doi:10.3390/met10010060_

Round 1

Reviewer 1 Report

The authors present a fascinating study of the different behaviours of selected metals in tension and compression. It will be expected to raise eyebrows among many of the engineering profession, and is therefore well worth publication.

There are, however, a number of relatively minor issues which would benefit from the authors’ attention.

The horizontal axis of both figures 3 and 4 are stated to be log strain. However, they appear to be extremely accurately linear.

The authors seem unaware of emerging recent thinking in the metallurgical world, in which the properties of metals are often found to be strongly influenced by the way they are melted and cast. This reviewer does not claim to be an expert in this new field.  It seems that most pouring operations can fold in the surface oxide on the liquid stream into the bulk metal. The folding action creates a double film which is called a bifilm. The bifilms act as cracks because there is little bonding across the oxide faces. In this way, different batches of metals are poured in different ways in different atmospheres (including vacuum of course, which still, I understand, contains sufficient oxygen to form bifilms) so that different batches of nominally the same alloy will contain different populations of cracks. If this is true, a dense crack population would result in different behavior for tension and compression. The authors will find much data on the web about these very thin and so nearly invisible defects, and their apparent ability to survive significant mechanical working. Much of this work seems centered on Birmingham University UK and North Florida University, USA.

Finally, the authors do not separate out a formal section ‘Conclusions’.  Many readers appreciate this feature, even though it will often appear unnecessary to the authors!  They are therefore encouraged to provide a ‘Conclusions’.

Author Response

Many thanks to the reviewer for the positive feedback and constructive comments.

A short feedback concerning axis in Fig. 3 & 4: The "logarithmic strain" is an alternative strain measure, in contrast to the "linear" Biot's strain. But both strain measures are plotted on a linear axis throughout the manuscript. This is why curves for nominal strain (Biot's strain; linear) are combined with curves for the converted log. strain in the same figure (to show the shift of the converted curve). The exact conversion is now provided (lines 65ff.). We apologize for this small confusion.

Thank you also for the inputs concerning processing (melting; casting; rolling and cold working). We know we could spend there even more time in providing additional explanations, why the observed yield asymmetry occurred. But also these would require an attempt to prove them, but as the project already was finished quite a long time ago, I cannot exert my colleagues in excess anymore.

We finally separated the conclusions from the quite long discussion section.

Reviewer 2 Report

The following comments are given by the reviewer (answers/discussions should be preferably included in the revised version of the paper):

- Abstract: More quantitative results should be added (the most significant ones) to highlight the scientific outcome of the paper.

- Introduction: Bullet points presenting the scientific contribution of the paper should be added at the end of this chapter.

- Page 2, Line 89 (and further): Reference Errors occur through the paper, please correct.

- Results: Typical fracture modes/surfaces (at least tension tests) may be added to highlight differences between the investigated materials.

- Figure 3: Vertical axes should be named as “True Stress”. Vertical axes is in some cases not well visible, please correct.

- Conclusions: Should be presented as a separate chapter/section including the bullet points with the most significant scientific outcome as well as quantitative results.

Author Response

Many thanks to the reviewer for the positive feedback and constructive comments.

We have tried to address the open points and suggested amendments for the text. Details are provided below:

The following comments are given by the reviewer (answers/discussions should be preferably included in the revised version of the paper):

- Abstract: More quantitative results should be added (the most significant ones) to highlight the scientific outcome of the paper.

The main focus of the paper is rather the verification of a new compression test method, and its application to biomedical high-strength alloys (with some unexpected observations). It was not primarily a new development or improvement of an alloy. Hence, absolute values and quantitative results are difficult to provide. The main proof of validity is maybe Table 1, where we might show the deviation of results from compression testing and from tensile testing. But we hope that the abstract nonetheless adequately transports our motivation and message.

- Introduction: Bullet points presenting the scientific contribution of the paper should be added at the end of this chapter.

We hope the rationale for the paper is adequately provided in the abstract and in the introduction, lines 70-77.

- Page 2, Line 89 (and further): Reference Errors occur through the paper, please correct.

Done (the error unfortunately occurred after uploading of the Word-doc, when converted to PDF)

- Results: Typical fracture modes/surfaces (at least tension tests) may be added to highlight differences between the investigated materials.

The tested samples are unfortunately not available anymore. Fractography was not really in the focus of the study.

- Figure 3: Vertical axes should be named as “True Stress”. Vertical axes is in some cases not well visible, please correct.

In figure 3 we have combined original curves (with nominal stress) together with their converted curves, which are 'True stress vs. log. strain', so we think it is simpler to just denominate the axis with "stress" (for both measures). The curves are labelled in the curve legends correspondingly, which should also help the reader.

- Conclusions: Should be presented as a separate chapter/section including the bullet points with the most significant scientific outcome as well as quantitative results.

Thank you for this hint. We have separated the Conclusions section from the Discussion.

Reviewer 3 Report

In the article the Authors perform compression tests and then compare the strain-stress curves with the tensile tests, with a view to obtain correct strain-stress relationship from first one. This idea is not new. The occurrence of additional forces is associated with friction forces. In addition, the stress – strain curve is distorted due to changes of a samples shape. There is also a huge number of works in which the stress-strain curves are analyzed for different strain-stess state (tensile, setting, torsion etc.). The reviewer believes that the Authors need to clearly show what exactly they offer to obtain the correct stress-strain curve. In its current form, the article is not of scientific interest. In addition, a significant number of random errors and problems must be corrected.

line 89 Error! Reference source not found.
line 133 Error! Reference source not found.
line 148 Error! Reference source not found.
Fig.3 Is it some reason to rotate the Fig?
line 165 -170 Is it the problem with a format
line 172 Error! Reference source not found.
line 202 Error! Reference source not found.
line 206 Error! Reference source not found.
line 212 Error! Reference source not found.
A conclusions should be added.

Author Response

Many thanks to the reviewer for his feedback on out manuscript.

The authors are aware that tensile and compressive testing were not recently invented. But what we had found out in the previous project is the fact, that compressive testing acc. to current standards is not really applicable for the determination of the flow curve during yielding of a ductile metal (our Ref. [1] in the present manuscript). The effect of friction was also investigated in detail in [1] (and it could be shown, that the drawback due to friction is minor in our suggested setup).

We hope we have provided enough details in the abstract and introduction to give the motivation and the rationale for the present manuscript. We manly focus on scientist applying FEM in applied R&D (biomedical engineering; implant development). This is also why the figures are rather large (and partly rotated, Fig. 3); they can easier be 'digitized' – something we also had done from Ref. [8].

All the formatting errors (for cross-references of Figures) should be corrected, they unfortunately occurred after uploading of the Word-doc, when converted to PDF.

Finally we have separated the Conclusions section from the Discussion – thank you for this hint.

Round 2

Reviewer 3 Report

Dear authors, thank you very much for your answers and corrections to improving the article. The reviewer recommends to publish the article after minor corrections of the introduction. Namely, both classical and modern works on this topic should be added and briefly discussed.

1. P.Ludwik, Elemente der technologischen mechanik, 1909
2. R. Hill: The Mathematical Theory of Plasticity, Oxford University Press, New York, NY, 1950, 355 pp.
3. Kulagin, R., Latypov, M.I., Kim, H.S. et al. Cross Flow During Twist Extrusion: Theory, Experiment, and Application. Metall and Mat Trans A 44, 3211–3220 (2013) doi:10.1007/s11661-013-1661-7
4. Chen, C., Beygelzimer, Y., Toth, L. S., et al., "Tensile Yield Strength of a Material Preprocessed by Simple Shear." ASME. J. Eng. Mater. Technol. 2016; 138(3): 031010. https://doi.org/10.1115/1.4033071

Author Response

We thank the reviewer for his feedback on our manuscript and the added suggestions for improvement.

The authors have tried to complement the paper by amending the Introduction of the manuscript, and by providing a better introduction to the topic. Unfortunately, the authors didn't have the first of the suggested references at hand, so referring to this book from 1909 would remain very vague. In addition, the process of first ordering the book and get familiar with it in order to correctly cite it would take too long for the value added to the present paper.

We were wondering if an overview of modern works can be provided by just two papers, which seem to originate from similar research groups, but we have added a reference to the latest paper about 'severe plastic deformation' (SPD).

We hope that the concerns of the reviewer are adequately addressed with this revision.

Suggested References for Introduction:

Ludwik P., Elemente der technologischen mechanik, 1909 Hill R.: The Mathematical Theory of Plasticity, Oxford University Press, New York,
NY, 1950, 355 pp. Kulagin, R.; Latypov, M.I.; Kim, H.S.; Varyukhin, V.; Beygelzimer, Yan
Cross Flow During Twist Extrusion: Theory, Experiment, and Application. Metall and Mat Trans A 44, 3211–3220 (2013), doi:10.1007/s11661-013-1661-7 Chen, C.; Beygelzimer, Y.; Toth, L.S.; Estrin, Y.; Kulagin, Roman
Tensile Yield Strength of a Material Preprocessed by Simple Shear, ASME. J. Eng. Mater. Technol. 2016; 138(3):031010. https://doi.org/10.1115/1.4033071